# Excess Zinc Alters Cell Wall Class III Peroxidase Activity and Flavonoid Content in the Maize Scutellum

**DOI:** 10.3390/plants10020197

**Published:** 2021-01-21

**Authors:** David Manuel Díaz-Pontones, José Isaac Corona-Carrillo, Carlos Herrera-Miranda, Sandra González

**Affiliations:** Laboratory for Tissue Biochemistry, Department of Health Sciences, Division of Biological and Health Sciences, Universidad Autónoma Metropolitana-Iztapalapa, Avenida San Rafael Atlixco No 186, Col Vicentina Iztapalapa, Ciudad de México CP 09340, Mexico; jicc@xanum.uam.mx (J.I.C.-C.); zeamais34@gmail.com (C.H.-M.); sandyglez@live.com (S.G.)

**Keywords:** activity index, germination, damage, tolerance limit, cell wall

## Abstract

Maize is one of the most important cereal crop species due to its uses for human and cattle nourishment, as well as its industrial use as a raw material. The yield and grain quality of maize depend on plant establishment, which starts with germination. Germination is dependent on embryo vigor and the stored reserves in the scutellum and endosperm. During germination, the scutellum epidermis changes and secretes enzymes and hormones into the endosperm. As a result, the hydrolysis products of the reserves and the different soluble nutrients are translocated to the scutellum through epithelial cells. Then, the reserves are directed to the embryo axis to sustain its growth. Therefore, the microenvironment surrounding the scutellum modulates its function. Zinc (Zn) is a micronutrient stored in the maize scutellum and endosperm; during imbibition, Zn from the endosperm is solubilized and mobilized towards the scutellum. During this process, Zn first becomes concentrated and interacts with cell wall charges, after which excess Zn is internalized in the vacuole. Currently, the effect of high Zn concentrations on the scutellum function and germinative processes are not known. In this paper, we show that, as a function of the concentration and time of exposure, Zn causes decreases in the radicle and plumule lengths and promotes the accumulation of reactive oxygen species (ROS) and flavonoids as well as changes in the activity of the cell wall Class III peroxidase (POD), which was quantified with guaiacol or catechin in the presence of H_2_O_2_. The relationship between the activity index or proportion of POD activity in the scutellum and the changes in the flavonoid concentration is proposed as a marker of stress and the state of vigor of the embryo.

## 1. Introduction

Maize, wheat, and rice are the agricultural products in greatest demand worldwide, occupying a large area of cultivation and production and having high economic benefits. Of these three crop species, maize occupies the largest planting area and is third in terms of economic importance [1,2]. Maize is used for human consumption or as fodder, and it is the basis for the production of processed foods and pharmaceutical products [1]. Highly productive Chalqueño maize is cultivated at an elevation of between 1800 and 2700 m in the high valleys of central Mexico [3]. This cultivar has high germination and vigor, a long life cycle, moderate resistance to drought in the middle stage of growth, and is adapted to growth in volcanic soil with a high mineral content [4].

The maize kernel is a simple, dry fruit with a single monocarpellary and indehiscent seed. The endosperm makes up approximately 85% of the mass of the mature grain [5] and contains reserve substances such as proteins and carbohydrates, in addition to large amounts of nutrients, including zinc [6,7]. The embryo (Figure 1a) consists of a single cotyledon or scutellum that communicates with the embryonic axis through the nodal plate and is surrounded by the endosperm [5,8,9]. Compared to that of wheat and rice, the maize scutellum is a larger structure [10] and represents 11% of the grain mass and approximately 90% of the embryo. During its development, it accumulates large amounts of reserve substances, such as phytate, lipids, and various micronutrients, including zinc [7,11,12], all of which are used during germination to reactivate the metabolism and growth of the embryo [13]. During germination, the scutellum undergoes morphological modifications to form an epithelium [14] that secretes enzymes and phytohormones and incorporates various nutrients from the endosperm [15,16,17,18,19]. These modifications are determinants of the growth and survival of the embryo, because the external supply of nutrients is limited, which holds true until the embryonic root develops, grows, and acquires the capability to absorb nutrients from the soil, which coincides with a decrease in internal reserves [20]. The scutellum is an ephemeral but indispensable structure, and once its functions are fulfilled, it undergoes programmed cell death; in small scutella, this occurs 10 days after germination [21,22].

Cereal grains possess significant mineral reserves for germination and seedling establishment. The micronutrients in the pericarp represent 8% to 12% of the total cumulative reserves, 60 to 80% are found in the endosperm, and the scutellum contributes 15% to 35%. Maize varieties have a high capacity to accumulate zinc deposits in the endosperm, between 3.6 and 4.3 µg endosperm^-1^, while the scutellum stores approximately 1.9 to 2.5 µg scutellum^-1^, so the ratio of the amount of zinc between the endosperm and scutellum fluctuates between 1.44 and 2.26 [11]. In maize, the time when zinc mobilization occurs is variable. In accumulator types, the concentration of this element decreases abruptly in the endosperm between 0 and 24 h of imbibition, while other varieties have a stable zinc reserve during the first 48 h, which decreases as imbibition progresses. Thus, the rate of mobilization of zinc from the endosperm to the embryo depends on the variety of maize, the capacity to store the ion, and the activity of the scutellum [11,12]. The epidermis of the scutellum is a critical structure during germination, since it receives and is subjected to increasing concentrations of zinc from the endosperm, which affects and modulates the physiological state of the scutellum.

Zinc is a transition metal whose abundance in plants is lower than that of iron. Zinc is present in six different classes of enzymes [23]. Its function and reactivity depend on the geometry and binding characteristics of the enzyme ligand to Zn^2+^, which composes the active catalytic, co-catalytic, or structural sites [24,25]. Among other functions, Zn^2+^ is a component of zinc fingers and transcription factors, and it can participate in interactions determining quaternary structures [26,27,28,29]. Zn^2+^ is a structural component of ribosomes and RNA polymerase [27], and it helps to maintain the integrity of cell walls and membranes [27,30,31,32]. Zn^2+^ is important in the development of plants and is part of fundamental metabolic processes such as photosynthesis [27,33,34,35] and carbohydrate metabolism [36].

Different cereals have different levels of sensitivity to zinc deficiency, as follows: maize ≈ rice > barley > rye ≈ oat [30]. Gramineae grown in acidic soils are less susceptible to high concentrations of zinc, which are toxic to most dicotyledons, and the inverse is observed in alkaline soils [37]. There is greater hypertolerance in species from the Poaceae, Caryophyllaceae, and Laminaceae families [38]. Excess Zn^2+^ generates a wide range of responses among species. The responses vary based on the toxicity threshold, exposure time, composition of the nutrient medium, and physiological state of the plant, among other factors [39]. In *Pisum sativum*, concentrations of zinc greater than 1 mM inhibit growth [40], while concentrations of 1 to 10 mM zinc in ryegrass retard growth, and a concentration of 50 mM ZnSO_4_ causes total inhibition [41]. The soaking of seeds in water with high concentrations of zinc is a practical way to increase the germination, growth, and development of seedlings into mature plants and subsequently enhance zinc accumulation in the seeds without having to precisely establish the tolerance thresholds [42,43].

Plants incorporate Zn^2+^ in the early stages of development and accumulate it in reserve organs, such as the scutellum and endosperm. The concentration of zinc in plant tissues is controlled by chelation in the cell walls and incorporation into vacuoles. The chelation of zinc in seeds regulates its concentration using phytate that has accumulated in the vacuoles [44,45]; this process affects the bioavailability and quality of food for human intake [46,47]. In total, 86% of the phosphorus in the scutellum is stored within phytate [48,49], which represents between 50% and 80% of the phosphorus in the seed [50,51,52]. During germination, free phosphate (Pi) is produced through the action of phytases and is used in metabolism [50,53,54]. This occurs during the first five days of imbibition, and the endogenous source of Pi depends on this process [55,56]. The high endogenous content of phosphorus in the seed improves the early development of cereal seedlings [57,58,59,60]. There is an interaction effect between the concentrations of phosphorus and zinc; under certain circumstances, one of these elements can become deficient, such that their interaction is antagonistic [61].

The solubilization and mobilization of nutrients from one organ to another causes nutrients to accumulate in a terminal organ, altering its homeostasis. If the physiological range is exceeded, a toxic effect is generated [62,63]. Concentrations of zinc above tolerable levels lead to an increase in the concentration of reactive oxygen species (ROS). If the concentration exceeds a critical level, an oxidizing state is generated during germination and seedling development [64,65,66,67]. To regulate stress, a complex system is induced to control excess ROS through the ascorbate–glutathione cycle [68,69,70], which is reinforced by a secondary antioxidant mechanism [71,72] involving flavonoids [73,74,75] and class III peroxidase (POD) [69,76,77,78]. In the germination phase, the secondary antioxidant system of flavonoids–class III peroxidase is critical [79,80] due to the low concentrations of ascorbate and reduced glutathione; however, the concentrations of these antioxidant metabolites increase in the scutellum, starting from three days after imbibition begins [81].

Therefore, during germination and early postgermination, the maize scutellum is subjected to a microenvironment in which the increase in the concentration of zinc from the endosperm affects/modulates the functionality of this organ. The response of the scutellum to exposure to high concentrations of zinc in conjunction with other nutrients, such as Pi, or organic molecules, such as citrate, is unknown. Likewise, its impact on the growth of the rest of the embryo is unknown. This article shows the negative effect of high concentrations of zinc on the growth of the embryo, the oxidative stress generated, the changes in the concentration of flavonoids, and the activity of class III POD in the scutellum during the germination to postgermination phase of maize. We propose the use of the ratio of POD activity to the flavonoid concentration as a stress marker to complement routinely used vigor tests.

## 2. Results

### 2.1. Germination and Growth

Maize embryos were imbibed in double-distilled water at a pH of 6.8 (H_2_O), double-distilled water at a pH of 4.5 (H_2_O+H^+^), 10 mM of potassium phosphate buffer at a pH of 4.5 (Pi), 10 mM of sodium citrate buffer at a pH of 4.5 (Cit), or experimental solutions of Pi or Cit buffer with ZnCl_2_ at concentrations of 1 mM (Zn1), 10 mM (Zn10), or 50 mM (Zn50).

After 24 h of imbibition, the germination rates of the embryos in the different imbibition media ranged from 83.3% to 100%, and the differences were not significant between groups (Figure 1b), except for Zn50 medium in which the germination rate was below 60%.

The radicle growth between 0 and 72 h (Figure 2a,b) was fit to a logarithmic regression (Appendix A). The radicle growth was similar under imbibition in H_2_O, H_2_O+H^+^, Pi, Cit, or Zn1 in both Pi and Cit buffers. The length at 48 h was 4.3 to 4.5 times larger than that at 24 h, while at 72 h, it was 6.6 to 7.8 times, larger than at 24 h (Figure 2a,b). A negative effect on growth was observed under imbibition in Zn10: the radicle length was 70 to 77% that of the control at 24 h and 65.6 to 67.1% at 72 h (Figure 2a,b). A drastic decrease in the size of the radicle was observed under Zn50, as the elongation was only 44.4% to 50% that of the control at 24 h and 16.4% at 72 h. 

Compared to that of the radicle, the growth of the plumule lagged over time (Figure 2c,d). Unlike the logarithmic model of the radicle, the mathematical model for the growth of the plumule was adjusted to a power regression (Appendix A). Plumule growth was similar for the embryos imbibed in H_2_O, H_2_O+H^+^, Pi, Cit, Zn1, and Zn10 in both buffers and was 3 to 3.8 times the 24 h level at 48 h and 6 to 7.6 times the 24 h level at 72 h (Figure 2c,d). The presence of Zn50 caused a decrease in the size of the plumule, reaching 31.8% or 38.0% that of the control at 48 h in Cit or Pi buffer, respectively, and 30.2% or 29.2%, respectively, at 72 h.

### 2.2. ROS Location

The effects on radicle and plumule growth were associated with an increased zinc concentration and correlated with the intensity of 3,3′-diaminobenzidine (DAB) staining in embryos after 24 h of imbibition (Figure 3). Germination in H_2_O caused slight staining in the expanding radicle, with increased intensity in the basal domains of the scutellum on the side that contacted the endosperm, the side attached to the embryonic root, and the upper-middle part of the confluence of the scutellum with the nodal plate of the embryonic axis (Figure 3b). The acidification of the imbibition medium intensified the staining: imbibition in H_2_O+^H+^, Pi, or Cit increased the reaction on the abaxial side of the scutellum (Figure 3c–e), which resulted in a basal staining pattern at 24 h. The presence of zinc increased the intensity of the staining depending on the concentration in the following order: Zn50 > Zn10 > Zn1. The intensity was lower in the presence of zinc in Pi vs. Cit (Figure 3d vs. Figure 3f–h vs. Figure 3e vs. Figure 3i–k).

### 2.3. POD Activity

The activity of Class III Peroxidase (POD) from the scutellum varied depending on the experimental conditions (Figure 4a,e and Figure 5a,e). The POD activity with catechin + H_2_O_2_ as substrates (POD-Cat) showed similar results under H_2_O, H_2_O+H^+^, Cit, and Pi. Compared with that at 24 h, the activity at 48 h increased 1.79–2.06 times and was similar between 48 and 72 h (Figure 4b,f). The presence of Zn1 yielded POD-Cat activity that fluctuated around the control level. It was slightly higher under Cit (Figure 4a–d) and lower in Pi (Figure 4e–h). At 24 h, the imbibition in Zn10 in both buffers induced a significant decrease of 23.2% to 33.1% in POD-Cat compared to in the control (Figure 4b,f). In contrast, the activity increased at 48 h compared to that at 24 h (Figure 4a,e), the increase of which was 4.22 or 5.02 times greater (with Cit+Zn10 or Pi+Zn10, respectively; Figure 4a,g). At 72 h, the activity increased differently depending on the buffer used (Figure 4d,h). At 24 h, Zn50 caused the greatest decrease in POD-Cat activity compared to the controls (Figure 4b,f). In the presence of this zinc concentration, POD-Cat increased 5.0–6.0 times at 48 h compared to the controls (Figure 4a,e), but was lower than that obtained under Zn10. The presence of Zn50 for 72 h induced the greatest activity level recorded in the study, an increase of 1.76 to 2.09 times compared to that at 48 h (in Cit or Pi, respectively; Figure 4a,e) and 2.3 to 2.6 times compared to that of the controls at this time (Figure 4d,h).

In the scutellum, the activity of POD with guaiacol + H_2_O_2_ (POD-Gua) was higher than the activity of POD-Cat, and the greatest increase occurred between 24 and 48 h (Figure 5a,e). At 24 h, the activity occurred in the following order: Cit ≅ Pi > H_2_O+H^+^ > H_2_O (Figure 5b,f). At 48 h, the activity level was similar between these conditions (Figure 5c,g), with an increase of 1.34 to 1.85-fold compared to the level at 24 h (Figure 5a,e). Compared with the level at 48 h, the activity level at 72 h was maintained or decreased slightly (Figure 5a,e,d,h). The presence of Zn1 for 24 h caused the POD-Gua in the scutellum to fluctuate around the level of the control (Figure 5b,f). After 48 h of imbibition, the increase in activity was 2.0 to 2.9 times that at 24 h. Compared with the level at 48 h, the activity at 72 h decreased under Cit+Zn1 or increased under Pi+Zn1 (Figure 5a,e). Imbibition of Zn10 for 24 h caused a slight decrease in the activity of POD-Gua compared to the control in Cit or Pi (Figure 5b,f). At 48 h, under Zn10, the activity level increased by ~2.5 times compared to that at 24 h (Figure 5a,b); at this time, the increase was 1.5 to 1.6 times that of the control, the values of which were similar to those achieved with Zn1. Compared to that at 48 h, the activity of POD-Gua at 72 h in the presence of Zn10 was approximately the same (Figure 5a,b), which was higher than that of the respective controls at the same time (Figure 5d,h). Imbibition of Zn50 caused a significant reduction in the activity of POD-Gua at 24 h, which resulted in the lowest value recorded in the study (Figure 5b,f). The increase in activity between 24 and 48 h varied from 2.3 to 2.5 fold in the presence of Zn50 in Cit or Pi, and the values stayed below those of Zn1 and Zn10 (Figure 5a,e). Compared with that at 48 h, the activity level at 72 h increased by 1.16 to 1.22 times, with an activity level at 72 h similar to that of Zn10 in both buffers (Figure 5d,h).

The relationship between the activities of POD-Cat and POD-Gua fits a logarithmic curve, in which the values of the activities of both substrates at short imbibition times grouped together, and after a prolonged period of time, the data dispersed, forming a tail-like pattern (Figure 6a,b). The dispersion observed was due to the different increases in the activity response between both substrates under the different experimental conditions; the dispersed values corresponded mainly to Zn50 and some values of Zn10 (Figure 6a). This allowed us to establish an activity index (AI), which varied based on the imbibition time, zinc concentration, pH, and composition of the imbibition medium (Figure 6c,d). At 24 h, an effect of pH on the AI was observed: the AI ranged from 7.08 to 5.9 under imbibition in H_2_O or H_2_O+H^+^ in both Cit and Pi. Imbibition under Cit+Zn1 caused an AI close to that of the Cit control and within the above range, with an increase in the AI proportional to the zinc concentration that was outside the above range for Cit+Zn10 and Cit+Zn50 (Figure 6c). The presence of zinc in Pi also affected the AI: Pi+Zn1 yielded a value below the values of the above interval, while Pi+Zn10 and Pi+Zn50 presented AIs higher than the above interval and greater than their counterparts in Cit (Figure 6d).

At 48 h of imbibition in H_2_O, H_2_O+H^+^, or Cit, the AI was 6.94 to 6.1, which was within the range observed at 24 h. The AI was lower in Pi. Imbibition in Cit+Zn1 maintained the AI within the interval found in the controls, but the AI decreased under Cit+Zn10 and Cit+Zn50 (Figure 6c, Appendix A). At this time, the AI values for Pi+Zn10 and Pi+Zn50 were below 4.4 (the Pi control value), while Pi+Zn1 maintained a value similar to that observed at 24 h (Figure 6d). At 72 h, under the different conditions, the AI was lower than the interval observed at 24 h, with values of 5.6 to 4.2 for the controls. Imbibition in Cit+Zn1 also yielded an AI within this range, but it was slightly lower than that of the Cit control. The AI decreased as the zinc concentration increased, presenting values equal to or less than 3.6 (Figure 6c). The scutellum imbibed in Pi+Zn1 had an AI greater than the values of the different intervals of the controls at different times (8.9 ± 0.5; Figure 6d). An increase in the zinc concentration in Pi led to AI values that were ≤ 3.6, and Pi+Zn50 had the lowest AI recorded in the study (Figure 6d).

### 2.4. Concentration of Phenolic Compounds

The concentration of phenols showed differences between the methods used (vanillin and DMACA methods) due to the specificity of the types of flavonoids with which they react. However, the relative changes were similar between the two techniques used, and a greater concentration of phenolic compounds was observed in the extracts with the vanillin method (Figure 7a,b) compared to the DMACA method (Figure 7c,d). The imbibition in Zn50 resulted in the hardening of the tissue, and with this, there was an artifact in the extraction that prevented precise quantification. 

The concentration of flavonoids in the scutellum quantified with the vanillin method (Figure 2a,b) was similar to the concentration in samples germinated in H_2_O, H_2_O+H+, Cit, and Pi at 24 h. The presence of Zn1 at this time caused a slight increase of 1.16 to 1.41-fold, while Zn10 increased by 1.1 to 1.36 fold compared with the concentrations in Pi or Cit controls, respectively. Imbibition for 48 h increased the concentration of flavonoids in the controls to between 2.29 to 3.0 times the amount present at 24 h.

The presence of Zn during 48 h of imbibition produced a greater increase in the flavonoid content compared with that of the controls, and this increase was proportionate to the ion concentration. The greatest level of accumulation occurred at 48 h and was caused by imbibition in Zn10, in which the flavonoid concentration increased by 1.65 times compared with the control, equal to an amount at 48 h that was ~3.5 times higher than that at 24 h. 

After 72 h of imbibition under the control conditions at a pH of 4.5, the concentration of flavonoids was slightly lower than that detected at 48 h. In the same way, germination at 72 h in Zn1 induced a concentration that fluctuated around that obtained at 48 h. Compared with that at 48 h, the concentration of flavonoids detected in the scutellum in Zn10 at 72 h decreased. The concentration was similar to that of Cit+Zn1 and was slightly lower than that of Pi+Zn1 for this imbibition time (Figure 7a,b).

The quantification of flavonoids in the scutellum by means of the DMACA method revealed changes similar to those of the vanillin method, although only a small fraction of the flavonoids reacted with DMACA (Figure 7c,d). At 24 h, the quantity fluctuated in the following order: Cit ≅ Pi > H_2_O+H^+^ > H_2_O. In addition, the presence of Zn1 and Zn10 at this time significantly increased the concentration compared with that of the control. The increase in flavonoids detected between 24 and 48 h of imbibition fluctuated between 1.27 and 1.34 times that of the control and Zn1 treatments and up to ~2.3 times that with Zn10, showing a smaller increase in the concentration detected compared with the amount detected via the vanillin method. The maximum concentration of flavonoids at this germination time was recorded under Zn10, the amount of which was ~1.87 times that of the control value (in the absence of zinc). At 72 h, the concentration of flavonoids was similar between the control conditions, with the value being lower than that found at 48 h. With Zn1, the amount was similar to that detected at 48 h for this treatment, and there was a decrease in flavonoids under Zn10, which was similar to that found with the alternative quantification method (Figure 7c,d). The imbibition in Zn50 resulted in the hardening of the tissue, and with this, there was an artifact in the extraction that prevented precise quantification.

### 2.5. Relationship between POD Activity and Flavonoids

The relationship between the concentration of flavan-3-ol and the activity of POD-Cat or POD-Gua showed an association in which the values were grouped mostly into two groups, with some values being outside these groups (Figure 8). The first group corresponded to the determination at 24 h of imbibition, at which the lowest level of POD activity and concentration of phenols were recorded. The second group corresponded to the determinations at 48 and 72 h under most of the experimental conditions tested, at which time the highest POD activity levels and concentrations of flavonoids were recorded. Outside this grouping were the data corresponding to the POD-Gua vs. flavan-3-ol relationship in the presence of Pi+Zn10 or Cit+Zn10 at 48 h and Pi+Zn1 at 72 h (Figure 8a,b). For the ratio of POD-Cat vs. flavan-3-ol, the dispersed values were associated with Pi+Zn10 or Cir+Zn10 at 48 and 72 h of imbibition (Figure 8c,d). The relationship between AI and flavan-3-ol showed dispersion at 24, 48, and 72 h under Pi+Zn10 and Cit+Zn10 and at 72 h under Pi+Zn1 (Figure 8e,f).

## 3. Discussion

### 3.1. Importance of the Scutellum in Maize

The scutellum represents approximately 90% of the embryo, an organ that is larger in maize than in other cereal crop species used for food [10]. The relationship between the size and the concentration of accumulated reserves in the scutellum is related to the supply of nutrients to the embryo axis, and this impacts the germination and growth of the seedling. The scutellum is the source of most carbon, nitrogen, and minerals, such as phosphate and zinc, making it a critical organ. Depletion of the scutellum reserves leads to metabolic and structural changes that prompt it to secrete enzymes to hydrolyze the reserves from the endosperm. Then, the products and soluble nutrients generated are transported by the scutellum and are finally transferred to the embryonic axis to maintain its growth. The efficiency with which this process is carried out affects the growth of the seedling and its ability to better acclimate to adverse environmental conditions which, in the long term, impacts plant vigor and the quantity and quality of the grain produced.

### 3.2. Zinc as a Nutrient and Its Disposal

Zinc is a micronutrient, and its proper mobilization and disposal are critical factors for seedling establishment. This element is stored in different concentrations between the endosperm and the scutellum, with a ratio of 1.44 to 2.26 between the two organs [11]. In Chalqueño maize grains, there is differential accumulation of zinc during germination, intense accumulation is observed in the epidermis of the scutellum and the endosperm that borders this organ (Appendix A). The scutellum participates in the mobilization of zinc from the endosperm for subsequent translocation to the embryonic axis [12]. The solubilization of zinc from the endosperm during imbibition generates an abrupt change in the concentration of this ion in the microenvironment that surrounds the scutellum, impacting the function of this organ. Therefore, the mobilization and accumulation of ions in the scutellum must be regulated to preserve the homeostasis of the organ and avoid the adverse effects of an increase in concentration. The amount of free zinc can be regulated first by its chelation in the cell wall, and excess ions are internalized and transported into the vacuoles, where they are chelated with free phosphates or phytate [45]. This type of interaction causes alterations to the solubility and bioavailability of both Pi and zinc, which can generate mild stress to severe damage in the structure, as observed with hardening in the presence of Zn50.

The availability of zinc, and therefore its effect, depend largely on the concentration of this soluble ion and the interactions that this ion has, among others, with phosphorus and organic acids that can bind to heavy metals, causing a deficiency in the soluble fraction of zinc [82,83,84]. Most studies involving zinc have used ZnSO_4_ instead of ZnCl_2_ as an external source of zinc, because under equimolar conditions, the latter is more toxic. However, during germination, SO^2-^ is beneficial as an essential macronutrient for development and growth, causing the effects of Zn^2+^ to be partially masked and thereby increasing the complexity of the experimental design and interpretation of the results. For this reason, we decided to use ZnCl_2_ to evaluate the response of the scutellum to this ion.

### 3.3. Effects of Zinc on Germination, Vigor, Growth, and the Redox State

The growth of the radicle and plumule during the first 72 h of imbibition was adjusted to fit different mathematical models. During the post-germination period, the radicle must develop and grow to incorporate water and nutrients into the seedling; in the case of cereals, the embryo radicle exhibits ephemeral growth (due to meristematic cells dividing, acquiring their identity and then differentiating). Due to these characteristics, the radicle growth was adjusted to a logarithmic model. However, the plumule grows and differentiates later than the radicle does, so the growth of the plumule was adjusted to fit a power regression model, as a constant degree of expansion of the leaf blades that constitute it was observed (Appendix A). Excess Zn^2+^ generates a wide range of responses that depend on the exposure time, toxicity threshold, medium composition, and physiological state of the plant [39]. The external application of ZnCl_2_ to embryos simulates the imbibition and solubilization of zinc from the endosperm, with subsequent alteration of the microenvironment of the epidermis of the scutellum, where Zn^2+^ accumulates and displaces other ions in the cell wall. This surplus can initially be translocated to the vacuole of the epidermis of the scutellum, which entails altering the endogenous concentration and tolerance levels.

Chalqueño maize showed a high rate of germination under the different control conditions analyzed. After 24 h of imbibition, the scutellum had a baseline ROS state that was in line with previous findings from our research group [14]. This state allows it to maintain adequate and efficient metabolism, with the appropriate transfer of nutrients to the radicle and growing plumule from germination to the post-germination stage [13].

The imbibition of maize embryos in the presence of 1 mM ZnCl_2_ was shown to be tolerable for short periods of time, in which the radicle and plumule growth was similar to that of the control. However, there was a small increase in the intensity of the reaction with DAB in the basal region of the scutellum, indicating an increase in oxidative stress, which induced changes in the AI after a longer period of time and in the relationship between the AI and the flavonoid concentration. The greatest negative effect was observed with Zn10 and Zn50, and these were different for Cit vs. Pi. While the presence of Zn10 did not affect germination, it did generate a moderate amount of ROS in the scutellum, which correlated with the decrease in radicle growth but did not affect the growth of the plumule. In contrast, Zn50 caused the greatest level of ROS accumulation in the scutellum, indicating more oxidative stress, which generated negative effects on the growth of both the radicle and the plumule; these effects were exacerbated as the exposure increased, and therefore, this concentration is highly toxic. The effects of the physiological state of the scutellum impacted the adequate growth of the embryonic axis, which depends on the functional efficiency of the scutellum as the source organ and the embryonic axis as a sink organ [13]. The alteration of the basal state of the scutellum by the increased ROS due to the accumulation and interaction of zinc depends on the concentration and time of exposure, which differentially affect the growth of the radicle and the plumule from what occurs naturally. By establishing temporary windows of vulnerability in the growth and development of the embryo, this study indicates that 0–72 h is a critical phase in seedling establishment.

### 3.4. Effects of Zinc on the Amount of Phenols and POD Activity

Zinc is a redox-inert element that is incapable of performing an oxidation–reduction reaction by itself [85]. However, it indirectly produces toxic oxygen species such as superoxide and hydrogen peroxide [86,87]. Excess zinc generates oxidative stress, which alters the antioxidant system and causes cellular damage [85,88,89,90]. Zinc promotes oxidative stress through mechanisms partially associated with the generation of charge transfer complexes with the production of °CH_3_, °OH, and quinhydrones due to the interaction of phenolic radicals with Zn^2+^ in the cell walls [90,91]. The present study confirms that maize scutella exposed to high concentrations of zinc accumulate ROS, and the increase in the intensity of DAB staining is correlated with the concentration used.

Under normal conditions, germination is characterized by an increase in the amount of ROS within a controllable interval [65] and is associated with antioxidant mechanisms. Among the antioxidants involved are flavonoids and antioxidant enzymes such as POD. Flavan-3-ol, either free or associated with the cell wall, plays an antioxidant role by reacting with ROS in situ, forming products that are polymerized into large apoplastic protocyanins or esterified to O or S groups of polyols, wall proteins, and/or other phenolic compounds in the cell wall [92]. Plants exposed to 1 to 5 mM of Zn^2+^ accumulate ROS and phenolic compounds as well as leading to increased activity of POD as well as other enzymes [90]. The increase in the activity of class III POD is one of the antioxidant mechanisms that control the amount of ROS through interaction with flavonoids at this stage [14]. The efficiency with which the concentration of flavonoids and the activity of POD are modulated determines the ability to control the oxidative stress generated, which impacts the degree of development and growth of the embryo into a seedling. An insufficient antioxidant response leads to the accumulation of ROS above the physiological or tolerance window, causing a negative effect that can be moderate to severe and that hinders the establishment of the seedling.

This work shows that the degree of growth of the embryo is influenced by the composition of the imbibition medium, the concentration of the elements that constitute it, the pH, and the exposure time, which affect the scutellum in its functional state and cause a state of stress, as discussed above. The imbibition of embryos in H_2_O or H_2_O+H^+^ for 24 h establishes a baseline state with respect to the concentration of flavonoids and POD activity in relation to the amount of ROS. Imbibition in Pi or Cit maintains these conditions with little variation, which allows the values of the relationship between flavonoids and the AI to be grouped within small ranges under these four conditions with short exposure times. Imbibition for 48 h under the control conditions was found to significantly increase the concentration of flavonoids and the activity of POD compared with the levels observed at 24 h. The existing relationship between flavonoids and the AI resulted in the grouping of values into a second set. At 72 h, the control conditions allowed us to maintain the maximal level of growth of the radicle and plumule with a scutellum in the basal state; under these conditions, the concentration of flavonoids and the POD activity with both Gua and Cat were similar to that achieved at 48 h, allowing us to include the values of the flavonoid-vs.-AI ratio within the second group.

While imbibition in Zn1 did not affect the growth rate of the radicle or the plumule, its presence for short periods caused a small increase in flavonoids and different levels of decrease in POD activity between the two substrates, which was reflected in the change in the AI. These changes were associated with a slight increase in ROS staining, which indicated a slight level of stress that was controllable. The response to stress persisted after 48 h in the form of a change in the AI, mainly in the Pi buffer, which was evident at 72 h, in which the AI-vs.-flavonoid ratio was outside that of the control group. Therefore, this concentration has negative effects as the imbibition time increases. The presence of zinc with Cit is more amenable than with Pi, probably because citrate can bind to heavy metals and alter the disposition of this element (chelating and removing it from solution or introducing it into the vacuole) [66,84,93], while Pi generates, together with endogenous phytate, harmful effects, such as the precipitation of compounds and the hardening of tissue (data not shown).

Imbibition of Zn10 at 24 h resulted in marked accumulation of ROS in the scutellum, in which the basal amount of flavonoids was maintained. However, there was a change in the activity of POD, which increased the AI to values higher than those of the controls and Zn1. The AI-vs.-flavonoid ratio was outside the set of values found in the controls and mainly affected the growth of the radicle but not that of the plumule. In terms of increased imbibition times, changes in the concentration of flavonoids and POD activity at 48 h and a decreased concentration of flavonoids and a differential increase in POD activity at 72 h resulted in a decrease in the AI to 3.6, which caused the AI-vs.-flavonoid ratio to be outside the control value grouping.

Imbibition with Zn50 caused a decrease in the growth of both the radicle and the plumule, a decrease in the activity of POD, and the highest AI at 24 h under all experimental conditions, and this was associated with the greatest accumulation of ROS in the scutellum. The activity of POD-Cat at 48 h under Zn50 was lower than that under Zn10 but was higher at 72 h, the time at which the maximum level of activity of the study was recorded. The different levels of change in activity between the two substrates at 72 h generated the lowest AI recorded in the study.

Therefore, at 24 and 72 h, the AI is an adequate parameter for indicating the degree of stress in the scutellum, and when the relationship between the flavonoids and the activity of POD or the AI is determined for each substrate mixture, the dispersion of values causes extreme conditions or high stress to be evident. Therefore, it is evident that Pi+Zn1 for 72 h or Zn10 in both buffers from 24 to 72 h are conditions that generate low to moderate levels of stress. The effects of Zn50 are excessive and cause significant oxidative stress in the scutellum with an exacerbated response. However, the activity of POD is insufficient, and residual growth of both the radicle and the plumule occurs. Therefore, the relationship between the activity of POD or the AI and the flavonoid concentration are indicators of the degree of stress, which can complement the techniques that determine the vigor and use MTT, thus strengthening the testing of plant stress.

### 3.5. Impact of the Study at the Food and Agronomic Levels

Approximately 40% of the global population suffers from malnutrition due to factors including iron and/or zinc deficiency in food [94,95], an insufficient quantity or quality of nutrients in the diet, and, in extreme cases, a lack of access to food. These changes have accompanied increases in poverty rates [96,97,98]. Deficiency of iron and zinc in food causes severe malnutrition, leading to a deterioration in human health, particularly in children and mothers, at the gestational and lactation stages [32,99,100].

At a global level, it is a priority to have foods rich in essential elements, including cereal grains, which are the main source of food for humans [32,101,102,103]. A high crop yield can be obtained from seeds with a high level of vigor that produce plants that are larger in size in the shortest possible time and have a strong ability to acclimate to environmental changes. These plants, in turn, produce seeds with high vigor and a high nutrient content. Among the different agronomic strategies used to solve this problem is biofortification through the use of zinc-rich fertilizers or nutropriming techniques, in which the seed is imbibed in nutrient solutions with zinc at millimolar concentrations for 12 to 24 h [42,43,104,105,106]. However, the repercussions of these treatments on the physiological state of maize embryos and the involvement of the scutellum during germination are unknown. Therefore, the present study provides knowledge about the consequences of the doses of zinc used in nutropriming and biofortification, which allows us to establish molecular markers for the degree of stress and damage produced from germination to the establishment of seedlings. This study contributes to the refinement of these techniques by establishing exposure times and the upper limit of the tolerable concentration, the effects of pH, and the effects of zinc in the presence of other nutrients, such as Pi, or organic molecules, such as Cit. In addition, this study directly shows the effects of zinc through the use of ZnCl_2_, which eliminates the side effects derived from the use of ZnSO_4_.

## 4. Material and Methods

### 4.1. Biological Material

Grains from Chalqueño creole maize cultivated in Valle de Chalco, Mexico (acquired from producers in the local seed market) were used. Mature grains were cleaned to remove debris, put in plastic airtight containers and stored in a controlled system at 7 °C and 40% relative humidity.

### 4.2. Germination

Embryos were obtained from mature grains by manual dissection and then superficially disinfected with a solution of 3% sodium hypochlorite for 15 min, followed by three washes in sterile water. Five embryos were included per germinator, and the scutellum was exposed to the different imbibition media: double-distilled water at a pH of 6.8 (H_2_O), double-distilled water at a pH of 4.5 (H_2_O+H^+^), 10 mM of potassium phosphate buffer at a pH of 4.5 (Pi), 10 mM of sodium citrate buffer at a pH of 4.5 (Cit), or experimental solutions containing Pi or Cit buffer with ZnCl_2_ at concentrations of 1 mM (Zn1), 10 mM (Zn10), or 50 mM (Zn50). Concentrated sterile solutions (pH 4.5) of Pi, Cit, and Zinc were made, and at the time of use, the relevant mixtures were made. In this study, ZnCl_2_ was used instead of ZnSO_4_ to eliminate the effects of sulfate as a nutrient as well as its participation in the regulation of the oxidation state/cell reduction [107,108,109]. Then, the germinators were incubated in the dark at 25 °C for 24, 48, or 72 h. At the end of the incubation period, the lengths of the radicle and plumule were measured, as was the number of germinated embryos.

### 4.3. ROS Location

The production of ROS, mainly H_2_O_2_, was detected in embryos after 24 h of imbibition. For each experimental condition, the embryos were immersed in an 0.1% solution of 3,3′-diaminobenzidine (DAB) dissolved in McIlvaine buffer at a pH of 6.0 (0.1 M citric acid:0.2 M Na_2_HPO_4_; 2.7:7.3) for 20 min at room temperature. The embryos were washed three times in distilled water, and a digital image was obtained to determine the degree of staining [110].

### 4.4. POD Activity in the Scutellum

From the embryos treated for each experimental condition, the scutella were obtained by manual dissection on an ice bed. The scutella were weighed and used to obtain crude extracts for enzymatic quantification. The scutella were homogenized in 100 mM of phosphate buffer (pH 6.8) and centrifuged at 10,000× *g* for 30 min at 4 °C. The POD activity level was determined using the following methods: (a) For POD-Cat, the reaction mixture consisted of 440 μL of 50 mM Pi buffer (pH 6.8), 20 μL of enzyme extract, and 440 μL of 20 mM (+)-catechin, to which 4 μL of 3% H_2_O_2_ was added to begin the reaction. The absorbance was recorded at 475 nm every minute for 5 min. (b) For POD-Gua, the reaction mixture consisted of 870 μL of 50 mM Pi buffer (pH 6.8), 10 μL of enzyme extract, and 10 μL of 1 M guaiacol (Gua), to which 9 μL of 3% H_2_O_2_ was added to begin the reaction. The absorbance was recorded at 475 nm every 20 s for 2 min [14]. The use of two different reaction mixtures was due to the large number of POD isoforms and allowed us to characterize the functions of different groups, for example, enzymes that participate in cell wall processes such as lignification (that use guaiacol or alcoholic phenols as substrates) or those associated with antioxidant mechanisms that use flavan-3-ol (such as catechin or its oligomers) as a substrate [14].

To determine the changes in isoenzyme activities that occurred during imbibition and were associated with the zinc effect, the proportion of POD activity between the two substrates (or activity index) used in the reaction mixture was calculated to show the relative changes in the isoenzyme group, which could be used as a stress marker. The activity index (AI) of POD in the scutellum was calculated for the different experimental conditions and imbibition times using the following equation:$${\mathrm{AI}}_{t}=\frac{\mathrm{activity}\;\mathrm{POD}-{\mathrm{Gua}}_{t}}{\mathrm{activity}\;\mathrm{POD}-{\mathrm{Cat}}_{z}}$$
where AI is the POD activity index at imbibition time t, POD-Gua_t_ is the activity of the POD determined with guaiacol (DO/min scutellum) at germination time t, and POD-Cat_t_ is the activity of the POD determined with catechin (DO/min scutellum) at germination time t.

### 4.5. Concentration of Phenolic Compounds in the Scutellum

The phenols from the scutella prepared under different experimental conditions and germination times were extracted by homogenization in a mixture of 1% HCl in 70% ethanol (5:17) with constant stirring overnight in an orbital shaker at 65 rpm at room temperature. The mixture was centrifuged at 2200× *g* for 20 min at 4 °C, and the supernatant was used for the determination of the total phenol content. The modified vanillin method was used [111,112], which involved incubation for 20 min at 30 °C. The reaction mixture consisted of 200 μL of the phenol sample with 1200 μL of 1% vanillin solution dissolved in methanol and 8% hydrochloric acid in methanol (1:1). After 20 min, the absorbance was measured at 500 nm, and the phenolic compound concentration was interpolated from a standard curve of (±)-catechin. The concentration of flavan-3-ol was determined from the same sample through a reaction with *p*-dimethylaminocinnamaldehyde (*p*-DMACA) [113,114]. Briefly, 1 mL of 0.1% DMACA solution (dissolved in ethanol:chloroform at a 1:1 ratio) and 950 μL of 1 N hydrochloric acid (diluted in absolute methanol) were added to 200 μL of the sample. The mixture was incubated for 15 min at room temperature, and its absorbance was recorded at 640 nm. The results were interpolated against a standard curve of (±)-catechin. The use of the two methods for quantification in this study was chosen because of the reactivity differences with different kinds of phenolics. Unlike the DMACA method, which identifies flavan-3-ol, the vanillin method can be used to detect a greater variety of phenolics [111,112,113,114]. 

### 4.6. Statistical Analysis

The results are expressed as the mean ± standard error. For the morphometric measurements, 40 to 60 independent data points were used. For the quantification of flavonoids and enzymatic activity, three to five independent tests were used, each with four scutella. A one-way analysis of variance was followed by the Tukey–Kramer multiple-comparison test at a significance level of *p* < 0.05. The analysis was performed with NCSS and PASS 2000 software (NCSS, LLC, USA).

## Figures and Tables

**Figure 1 plants-10-00197-f001:**
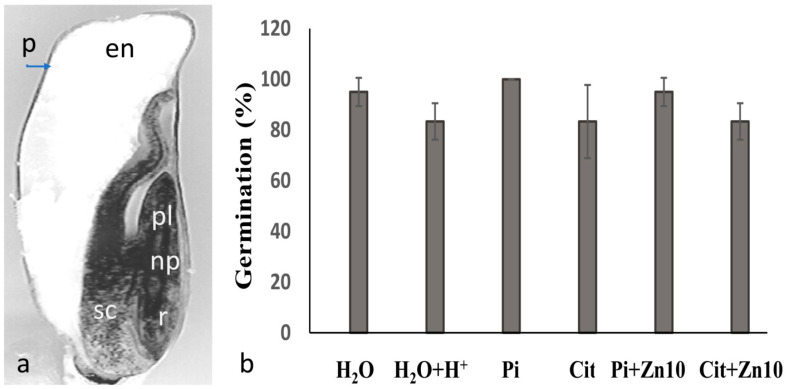
Grain structure and effect of zinc on germination. (**a**). Structure of maize grain. (**b**). Germination was determined in maize embryos imbibed for 24 h in the dark at a temperature of 25 °C in double-distilled water at a pH of 6.8 (H_2_O), double-distilled water at a pH of 4.5 (H_2_O+H^+^), 10 mM of potassium phosphate buffer at a pH of 4.5 (Pi), 10 mM of sodium citrate buffer at a pH of 4.5 (Cit), or Pi or Cit buffer with ZnCl_2_ at concentrations of 1 mM (Zn1), 10 mM (Zn10). The data represent the mean ± SE of n = 40 to 60 independent data points. ANOVA was performed, followed by the Tukey–Kramer multiple comparison test, with *p* < 0.05 representing significance. There was no significant effect on the percentage of germination among the experimental conditions shown. Symbols: en, endosperm, np, nodal plate; p, pericarp, pl, plumule; r, radicle.

**Figure 2 plants-10-00197-f002:**
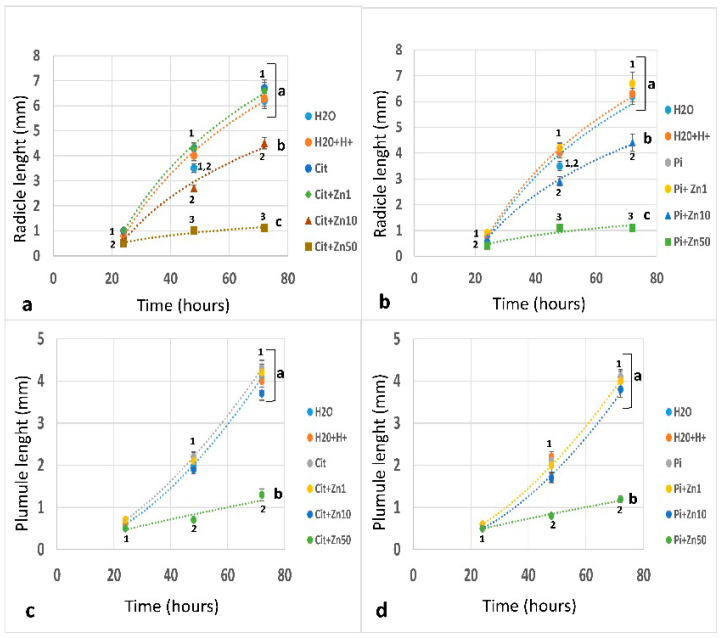
Effect of zinc on radicle and plumule growth. Maize embryos imbibed for 24, 48, or 72 h. (**a**,**c**): Effect of zinc at 1, 10, or 50 mM in the presence of 10 mM of Cit. (**b**,**d**): Effects of zinc at 1, 10, or 50 mM in the presence of 10 mM of Pi. There was a negative and significant effect on the radicle growth of embryos imbibed in Zn10 or Zn50 compared with the other conditions, while the expansion of the plumule was only affected by imbibition under Zn50. The data represent the mean ± SE of n = 40 to 60 independent data points. ANOVA was performed, followed by the Tukey–Kramer multiple comparison test, with *p* < 0.05 representing significance. Significant changes at each imbibition time for the different experimental conditions are indicated with different letters. Significant differences between experimental conditions during the imbibition interval (24 to 72 h) are indicated with lowercase letters.

**Figure 3 plants-10-00197-f003:**
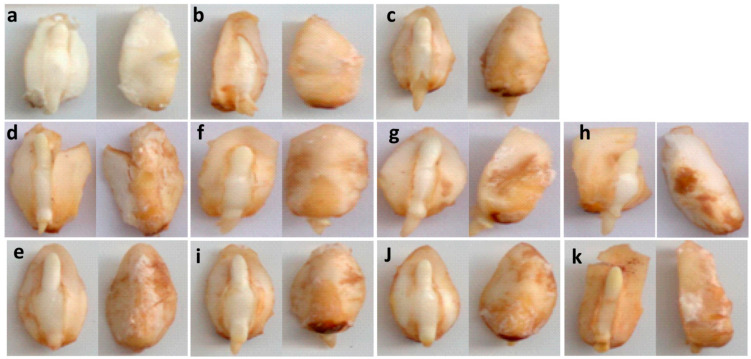
In situ location of reactive oxygen species (ROS). 3,3′-Diaminobenzidine (DAB) staining for the detection of mainly H_2_O_2_ in embryos imbibed for 24 h in different media. (**a**), Unstained embryo; (**b**), H_2_O at pH 6.8; (**c**), H_2_O+H^+^ at pH 4.5; (**d**), Pi; (**e**), Cit; (**f**), Pi+Zn1; (**g**), Pi+Zn10; (**h**), Pi+Zn50; (**i**), Cit+Zn1; (**j**), Cit+Zn10; and (**k**), Cit+Zn50.

**Figure 4 plants-10-00197-f004:**
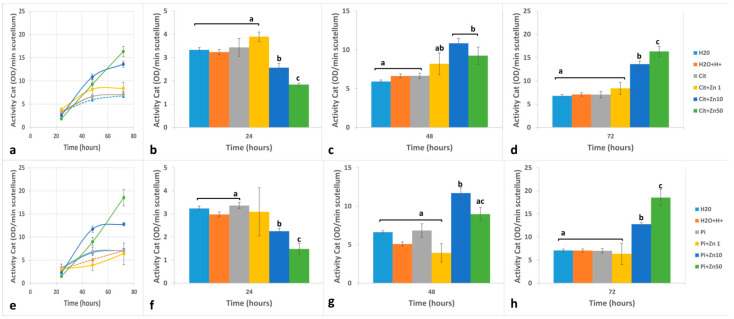
Effects of zinc on the activity of Peroxidase quantified with catechin + H_2_O_2_ as substrates (POD-Cat) from maize scutellum. Embryos were imbibed for 24 to 72 h in H_2_O, H_2_O+H^+^, Cit, Cit+Zn1, Cit+Zn10, Cit+Zn50, Pi, Pi+Zn1, Pi+Zn10, or Pi+Zn50. Activity was determined using catechin + H_2_O_2_ as substrates in (**a**): Cit or (**e**): Pi. Imbibition times were (**b**,**f**): 24 h; (**c**,**g**): 48 h; and (**d**,**h**): 72 h, in Cit or Pi, respectively. The data represent the mean ± SE of n = 3 to 6 independent assays. ANOVA was performed, followed by the Tukey–Kramer multiple comparison test, with *p* < 0.05 representing significance. Letters that are the same indicate similar variations, and the different letters indicate significant changes at each imbibition time.

**Figure 5 plants-10-00197-f005:**
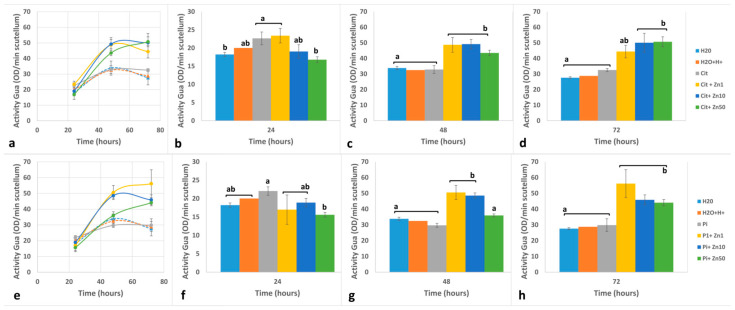
Effect of zinc on the activity of peroxidase quantified with guaiacol + H_2_O_2_ as substrates (POD-Gua) from maize scutellum. Embryos were imbibed for 24 to 72 h in H_2_O, H_2_O+H+, Cit, Cit+Zn1, Cit+Zn10, Cit+Zn50, Pi, Pi+Zn1, Pi+Zn10, or Pi+Zn50, and activity was determined using guaiacol + H_2_O_2_ as substrates. (**a**): Cit; (**e**): Pi. Imbibition times were (**b**,**f**): 24 h; (**c,g**): 48 h; (**d,h**): 72 h, in Cit or Pi, respectively. The data represent the mean ± SE of n = 3 to 6 independent assays. ANOVA was performed, followed by the Tukey–Kramer multiple comparison test, with *p* < 0.05 representing significance. Letters that are the same indicate similar variations, and different letters indicate significant differences at each imbibition time.

**Figure 6 plants-10-00197-f006:**
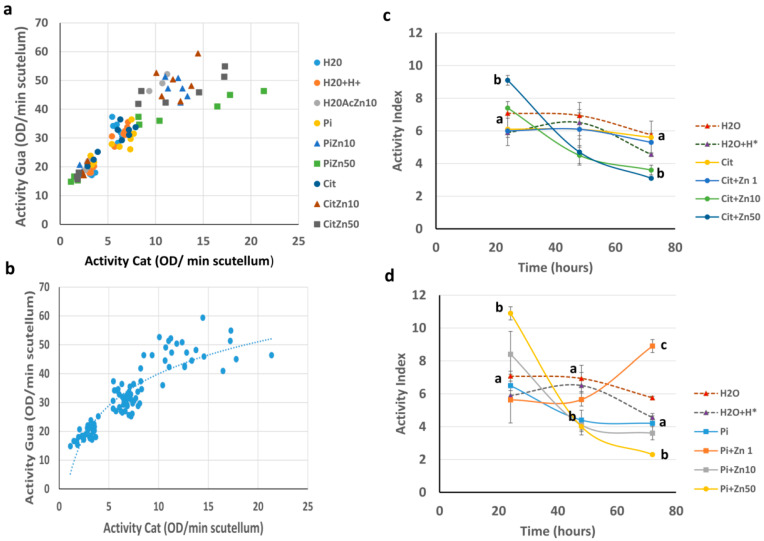
Peroxidase activity index in scutellum. (**a**) Relationship between the activity of POD-Cat vs. POD-Gua for the different experimental conditions during germination. (**b**) The dotted line indicates the regression model, which was fitted to a logarithmic relationship with the equation y = 15.933 ln (x) + 3.2831, with R² = 0.8114. (**c**,**d**) Relationship between activity index (AI) values for the activity of POD-Gua/activity of POD-Cat for each experimental condition during the germination. (**c**), ZnCl_2_ in citrate buffer; and (**d**), ZnCl_2_ in phosphate buffer. The graph shows the mean ± SE with 3 to 6 independent assays. ANOVA was performed, followed by the Tukey–Kramer multiple comparison test, with *p* < 0.05 representing significance. Letters that are the same indicate similar variations, and different letters indicate significant differences at each imbibition time.

**Figure 7 plants-10-00197-f007:**
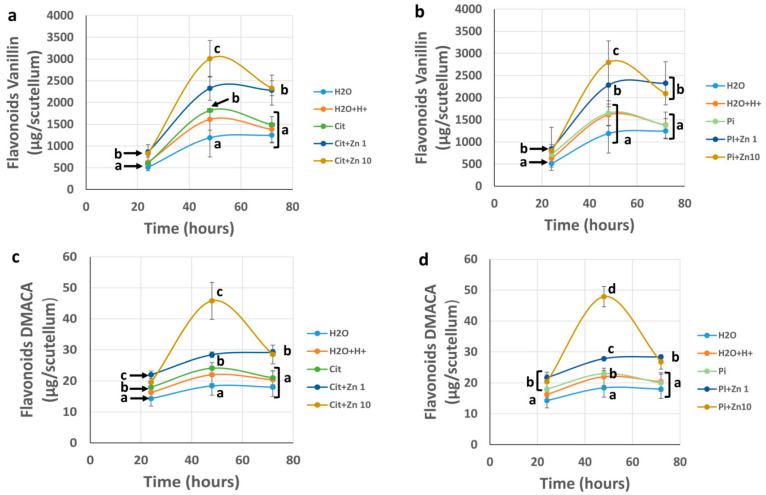
Concentration of phenolic compounds in the scutellum during germination. The concentration of phenols in the scutellum from embryos imbibed for 24, 48, or 72 h in H_2_O, H_2_O+H^+^, Pi, Cit, or Zn1, Zn10, or Zn50 dissolved in Cit or Pi buffer quantified (**a**,**b**): with the vanillin method; (**c**,**d**): with the DMACA method. ANOVA was performed, followed by the Tukey–Kramer multiple comparison test, with *p* < 0.05 representing significance. The different lowercase letters imply significant differences among experimental conditions for specific imbibition times. The data represent the mean ± SE of 3 to 6 independent assays.

**Figure 8 plants-10-00197-f008:**
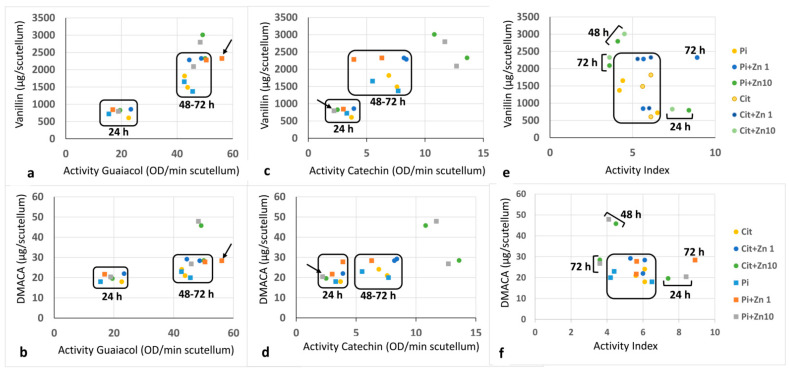
Relationship between peroxidase activity and the concentration of flavonoids. The association of POD-Gua, POD-Cat, or the activity index (AI) with the concentration of flavonoids quantified with the vanillin or DMACA method is shown for the different imbibition times and media. The grouping of values for the experimental conditions is shown in boxes, and the values dispersed from this grouping are indicated, along with the experimental conditions and imbibition times to which they belong. (**a**,**c**,**e**): Quantification of flavonoids with the vanillin method. (**b**,**d**,**f**): Quantification of flavonoids with the DMACA method. (**a**,**b**), Relationship between flavonoids and POD-Gua; (**c**,**d**): Relationship of flavonoids with POD-Cat; (**e**,**f**): Relationship of the concentration of flavonoids with the AI.

## Data Availability

Data sharing is not applicable to this article.

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
