# Peer review of "Excess Zinc Alters Cell Wall Class III Peroxidase Activity and Flavonoid Content in the Maize Scutellum"

_plants, 2021, doi:10.3390/plants10020197_

Round 1
Reviewer 1 Report
The review on the publication by Diaz-Pontones at al. under the title ‘Zinc Induces Cell Wall Stress and Changes in Class III Peroxidase Activity and Flavonoid Concentration’. The publication is well prepared.
I have only minor comments to the manuscript:
Line 31-33 I don’t think this is the proper place for abbreviations. According to the journal requirements, it has to be at the end of the manuscript.
I think authors have to add the general scheme of the maize seed at the beginning of the results part. It will be much easier for the reader the understand which part the authors mean in the article. I think it significantly improves the quality and what is a more critical understanding of the manuscript.
Reviewer 2 Report
Summary
The article describes the effect of treatments with exogenous Zn added in growth, oxygen radical production and location, peroxidase activity, amount of phenolic compounds and the relationships between the later two aspects in maize scutellum.
General comment
The article presents several drawbacks that make it unsuitable for publication. First, the introduction is long and contains many references, but key information is missing. Second, information is repeated in different sections. For example, in the two supplementary documents provided or between the introduction and results sections and the discussion. The results are presented in a confuse form.
Comments by section
Introduction
key information and references are lacking. For example, in the following sentence:
Maize varieties have a high capacity to accumulate zinc deposits in the endosperm, between3.6 and 4.3 μg, while the scutellum stores approximately 1.9 to 2.5 μg, so the ratio of the amount of zinc between the endosperm and scutellum fluctuates between 1.44 and 2.26
The information given is of little utility because it does not mention the concentration of Zinc in these tissues either in molarity or in ppm; neither the comparison with other values in other plant species.
A reference is missing for the following sentence:
“The sowing of seeds in high concentrations of zinc is a practical way to increase the germination, growth, and development of the seedling into a plant and subsequently enhance zinc accumulation in the seeds without having established the tolerance thresholds precisely.”
Results
Two supplementary files have been provided, one in Word and a PDF, containing the same information. The reason for giving the regression models is not presented, and their purpose is not mentioned. Supplementary Table 1 is not justified. The utility of these regression models has to be proven. The basic measurements are not shown.
Subscripts are incorrectly placed in Figure 1.
The reason to measure POD activity under two alternative conditions is not explained (catechin+H2O2 (POD-Cat); POD with guaiacol+H2O2 (POD-Gua); The biological relevance of AI is not explained .
Similarly, the reasons for using two methods of determination of flavonoids (vanillin and DMACA) are not given, and the differences between these two methods are also not mentioned nor discussed. The results of these experiments are described in a confuse way with some repetitions. It is disappointing that similar treatments are represented with different colours in figures 7a,c and 7b,d.
Discussion
The discussion is too long and contains aspects pertaining and/or repeated with/to the introduction as well as information from the results.
Overall, the article is long and confusing and makes difficult to see the biological relevance of the results.
